# Identification of Metastatic Lymph Nodes Using Indocyanine Green Fluorescence Imaging

**DOI:** 10.3390/cancers15071964

**Published:** 2023-03-25

**Authors:** Kyungsu Kim, Kook Nam Han, Byeong Hyeon Choi, Jiyun Rho, Jun Hee Lee, Jae Seon Eo, Chungyeul Kim, Beop-Min Kim, Ok Hwa Jeon, Hyun Koo Kim

**Affiliations:** 1Department of Thoracic and Cardiovascular Surgery, Korea University Guro Hospital, Korea University College of Medicine, Seoul 08308, Republic of Korea; 2Department of Biomedical Sciences, Korea University College of Medicine, Seoul 02841, Republic of Korea; 3Department of Thoracic and Cardiovascular Surgery, Chung-Ang University Gwangmyeong Hospital, Gwangmyeong 14353, Republic of Korea; 4Department of Nuclear Medicine, Korea University Guro Hospital, Korea University College of Medicine, Seoul 08308, Republic of Korea; 5Department of Pathology, Korea University Guro Hospital, Korea University College of Medicine, Seoul 08308, Republic of Korea; 6Department of Biomedical Engineering, Korea University College of Health Science, Seoul 02841, Republic of Korea; 7Interdisciplinary Program in Precision Public Health, Korea University, Seoul 02841, Republic of Korea

**Keywords:** indocyanine green, fluorescence imaging, metastatic lymph node, primary cancer

## Abstract

**Simple Summary:**

Indocyanine green (ICG)-based fluorescence imaging has been used to detect several types of tumors; however, its ability to detect metastatic lymph nodes (LNs) remains unclear. The feasibility of ICG for detecting tumors and metastatic LNs was evaluated in patients with lung or esophageal cancer, detected with computed tomography (CT) or positron-emission tomography (PET)/CT, and scheduled to undergo surgical resection. ICG-based fluorescence imaging could identify metastatic lymph nodes and enable surgeons to dissect them, potentially preventing cancer recurrence. However, the feasibility of ICG-based intraoperative detection of metastatic lymph nodes needs to be validated with further studies.

**Abstract:**

Indocyanine green (ICG) has been used to detect several types of tumors; however, its ability to detect metastatic lymph nodes (LNs) remains unclear. Our goal was to determine the feasibility of ICG in detecting metastatic LNs. We established a mouse model and evaluated the potential of ICG. The feasibility of detecting metastatic LNs was also evaluated in patients with lung or esophageal cancer, detected with computed tomography (CT) or positron-emission tomography (PET)/CT, and scheduled to undergo surgical resection. Tumors and metastatic LNs were successfully detected in the mice. In the clinical study, the efficacy of ICG was evaluated in 15 tumors and fifty-four LNs with suspected metastasis or anatomically key regional LNs. All 15 tumors were successfully detected. Among the fifty-four LNs, eleven were pathologically confirmed to have metastasis; all eleven were detected in ICG fluorescence imaging, with five in CT and seven in PET/CT. Furthermore, thirty-four LNs with no signals were pathologically confirmed as nonmetastatic. Intravenous injection of ICG may be a useful tool to detect metastatic LNs and tumors. However, ICG is not a targeting agent, and its relatively low fluorescence makes it difficult to use to detect tumors in vivo. Therefore, further studies are needed to develop contrast agents and devices that produce increased fluorescence signals.

## 1. Introduction

Lung and esophageal cancers are thoracic cancers with high mortality rates worldwide [1]. Surgery is typically considered an effective treatment strategy for early-stage thoracic cancers and can improve survival rates [2,3,4,5]. However, if patients with primary lung or esophageal cancer exhibit lymph node (LN) metastasis, the 5-year survival rate is significantly reduced for them [6,7]. Lymphadenectomy can increase the 5-year survival rate of patients with lung cancer from 42% to 58% and that of patients with esophageal cancer from 38.3 to 55% [8]. Therefore, precise detection and dissection of metastatic LNs is critical for improving survival rates.

Identification of metastatic LNs remains challenging, with occult LN metastasis reported in 8–11% of primary lung cancers [9,10] and 26–56% of esophageal cancers [11,12]. Extensive resection of cancer and locoregional LNs is the standard surgical approach for thoracic cancer surgery to avoid missing occult metastasis [13]. However, extensive dissection is often accompanied by an incidence of postoperative complications, such as recurrent nerve paralysis, chylothorax, and atrial fibrillation, which can eventually reduce quality of life [14]. Minimal resection may lead to metastatic LNs being overlooked, thereby causing cancer recurrence [15]. Therefore, it is important to identify and dissect metastatic LNs selectively and accurately during surgery.

In addition to primary lung cancer, the lung is a common site of malignant tumor metastasis [16]. Surgical resection is a feasible treatment option for metastatic lung cancer with a limited number of metastases and sufficient pulmonary function [17]. LN metastasis can also significantly reduce the survival rate of patients with metastatic lung cancer. The 5-year survival rate is 58.3% in the absence of LN metastasis and 24.8% with metastasis [18]. However, LN assessment is not routinely performed for patients with lung metastasis [19]. Only suspicious LNs are dissected during lung metastasectomy [20]. Considering the 12–19% incidence of metastasis in mediastinal LNs [17], selective resection of metastatic LNs is also important.

Morton and Cochran first introduced the sentinel LN (SLN) concept for detecting metastatic LNs in 2004 [21]. Currently, SLN biopsy is considered the standard surgical treatment for patients with melanoma and breast cancer [22]. However, its accuracy and feasibility in thoracic cancer need to be determined [23]. In addition, SLN biopsy has major limitations. First, after SLN dissection, it is necessary to wait for pathology results to confirm SLN metastasis, which may delay surgery [24]. Second, owing to skipped metastasis, SLN biopsy can lead to false-negative results. In a meta-analysis, skipped metastasis was reportedly found in 20–40% of patients with lung cancer [25,26] and 20–76% of patients with esophageal cancer [27,28]. Therefore, it is necessary to develop a novel method to directly detect metastatic LNs without SLN biopsies.

Positron-emission tomography (PET)/computed tomography (CT) are the most widely used methods for evaluating LN status [29]. According to several reports, the sensitivity of PET/CT for detecting metastatic LNs ranges from 64 to 80% while the specificity ranges from 70 to 100%, with a diagnostic accuracy of 73–79% [30,31,32]. The reasons for low preoperative diagnostic accuracy are as follows: first, because of inflammation, some enlarged LNs are easily misdiagnosed as metastatic LNs, leading to false-positive results. Second, some metastatic LNs are normal in size, leading to false-negative results. Moreover, intraoperative identification of metastatic LNs is difficult. Therefore, a new tool could accurately detect metastatic LNs and provide surgeons with information.

Since 2009, near-infrared (NIR) fluorescence imaging using intravenous injection of indocyanine green (ICG) has been used for intraoperative detection of hepatocellular carcinoma [33], and this method was subsequently employed to detect several types of tumors [34,35]. We previously reported that lung tumors could be intraoperatively detected using intravenous injections of ICG (1 mg/kg) [5]. However, the efficacy of ICG in detecting metastatic LNs has not been reported. 

In this study, we aimed to evaluate the ability of intravenous ICG injection to detect metastatic LNs and tumors in a mouse model of metastatic LNs. In addition, the clinical feasibility of this method was evaluated through performing ICG-based metastatic LN detection in patients with lung or esophageal cancer.

## 2. Materials and Methods

### 2.1. Cell Line

In this study, Lewis lung cancer cells labeled with green fluorescent protein (LLC-GFP) were used. The cell line was cultured in Dulbecco’s modified Eagle’s medium supplemented with 10% fetal bovine serum (Thermo Fisher Scientific, Grand Island, NY, USA) and 1% antimycotic (Thermo Fisher Scientific, Grand Island, NY, USA) at 37 °C in a 5% CO_2_ incubator under saturated humidity.

### 2.2. Preoperative Imaging 

Chest computed tomography (CT) and ^18^F-FDG positron-emission tomography (PET)/CT were performed on all 15 patients. For the chest CT, 100 mL of iopromide (ProsureM300; LG Life Sciences, Seoul, Korea) at 4 mL/s was administered and a 128-slice, dual-source CT scanner (Siemens Healthcare, Forchheim, Germany) was used for detection. CT images were obtained from all patients and stored on a picture-archiving and communication system (PiViewSTAR; Infinitt, Seoul, Korea) at a 3 mm section thickness. Two radiologists independently reviewed the images. The ^18^F-FDG PET/CT was performed using Gemini TF/16 channel PET/CT scanners (Philips Medical Systems, Cleveland, OH, USA). The patients were instructed to fast for at least 6 h to ensure that their blood glucose levels were ˂140 mg/dL before intravenous ^18^F-FDG administration. Radiotracer (6.0 MBq/kg) ^18^F-FDG was intravenously administered, and scanning was performed after 60 min. PET images were reconstructed using an iterative reconstruction algorithm based on the CT images for attenuation correction. The images were reviewed on an interactive video display provided by MIRADA XD3 software (MIRADA Medical, Oxford, UK). The standardized uptake value (SUVmax) was used to quantitatively determine the FDG activity. TLNs with abnormal FDG uptake (SUVmax > 2.5) were considered positive [29]. All images were analyzed by two experienced nuclear-medicine physicians.

### 2.3. Animal Model and Near-Infrared (NIR) Fluorescence Imaging

All study procedures, including animal care and handling, were approved by the Institutional Animal Care and Use Committee of Korea University (KOREA-2020-0110). C57BL/6 mice (age: 6 weeks; weight: 20–25 g; Orient Biotech, Seoul, Korea) were used (1-week adaptation). Ten mice were divided into two groups: control (n = 5) and footpad tumor model with LN metastasis (n = 5). LLC-GFP cells (5 × 10^5^) were injected into the footpads of the mice, and a footpad tumor model was established after 3 weeks of injection. ICG (25 mg vials; JEIL, Seoul, Korea) was dissolved in 10 mL of an injectable solution to yield a 2.5 mg/mL (3.2 mM) stock solution, which was intravenously injected at doses of 2 mg/kg via the tail veins of the mice. After 12 h, ICG distribution in the tumors and LNs was detected using a Davinci-Invivo^TM^ imaging system (Davinch-k, Seoul, Korea) at excitation and emission wavelengths of 769 and 809 nm, respectively, following which resection was performed. The GFP fluorescence signal was detected in the frozen section slides at excitation and emission wavelengths of 475 and 520 nm, respectively. Further, hematoxylin and eosin staining were performed for histological analysis.

### 2.4. Patients and Preoperative Imaging

This study was approved by the Institutional Review Board of Korea University Guro Hospital (2020GR0181). Between May 2017 and March 2019, 15 patients who were preoperatively diagnosed with lung or esophageal cancer and scheduled to undergo curative surgery were enrolled. An ICG dose of 2 mg/kg was administered intravenously to each patient with cancer at 12 h before surgery. Fluorescence images of the patient specimens were obtained using the Davinci-Invivo™ imaging system mentioned above. Patients with liver dysfunction (aspartate transaminase, alanine transaminase >2.5 times the normal value), hypersensitivity, or adverse reactions to ICG and those receiving neoadjuvant chemotherapy were excluded.

### 2.5. Surgical Procedures

#### 2.5.1. Lung Cancer

Surgical approaches were discussed individually for all patients and were based on cancer characteristics (localization, size, and metastasis) or the surgeon’s preference. Minimally invasive approaches, such as video-assisted thoracoscopic surgery (VATS) or robotic lobectomy, were the preferred options for primary lung cancer; however, if a patient had pulmonary dysfunction, wedge resection was selected. In this study, except for one patient who had a small-sized (0.7 cm) tumor with poor pulmonary function and who underwent wedge resection and dissection of one suspicious LN, all patients with primary lung cancer underwent VATS or robotic lobectomy with systematic mediastinal LN dissection. 

For metastatic lung cancer, when the lesion was located in the peripheral lung and there was no other evidence of lung metastasis, minimally invasive surgery, such as VATS wedge resection, was performed. LN dissection was performed when metastasis was suspected based on preoperative CT or PET/CT, or if a LN was found to be enlarged during surgery.

#### 2.5.2. Esophageal Cancer

VATS is the standard surgical approach for esophageal cancer [36]. In patients who had middle and lower thoracic esophageal cancer, esophagectomy with 2-field LN dissection was performed. 

After resection of the cancer and the LNs, ICG fluorescence imaging of the tumors and LNs that were enlarged, had suspected metastasis, or were anatomically key regional LNs of the lung or esophagus were obtained using an NIR fluorescence-imaging system. ICG fluorescence signals of the tumors and LNs were analyzed using the signal-to-background ratio (SBR). All the specimens were subjected to histological examination. Pathologic staging of the lung cancer was based on the World Health Organization histological classification and the Union for International Cancer Control (UICC, 8th edition) TNM classification. Esophageal cancer was also classified according to the 8th edition of the American Joint Committee on Cancer/UICC TNM classification.

### 2.6. Statistical Analysis 

One-way analysis of variance was used to analyze the differences in SBRs between tumors, metastatic LNs, and normal LNs. All graphs, calculations, and statistical analyses were performed using GraphPad Prism software, version 8.4.3, for Windows (GraphPad Software, San Diego, CA, USA). *P*-values were considered statistically significant at P < 0.05. Diagnoses of metastatic LNs based on ICG fluorescence imaging, CT, and PET/CT were compared with those based on histopathological findings. Sensitivity, specificity, positive predictive values (PPVs), negative predictive values (NPVs), and accuracy were determined using the fourfold table diagnostic test.

## 3. Results

### 3.1. Identification of Tumors and Metastatic LNs Using ICG Fluorescence Imaging in the Mouse Model

Three weeks after LLC cell injection, LN metastasis was successfully established in the footpad-tumor-model mice, as confirmed with GFP signals and pathology (Figure 1). We compared the ability of ICG to detect normal and metastatic LNs in mice. Figure 1 shows that the NIR fluorescence signals of the tumors and the metastatic LNs in the footpad tumor model were significantly higher than those in the normal mice. In addition, histology and fluorescence microscopy revealed that the ICG was distributed to GFP-labeled cancer cells in metastatic popliteal LNs (Figure 1). The SBRs of tumors (2.8 ± 0.5) and metastatic LNs (1.9 ± 0.2) were significantly higher than that of normal LNs (1.1 ± 0.1) (tumors vs. normal LNs, *p* < 0.0001; metastatic LNs vs. normal LNs, *p* < 0.0001). There was no significant difference in SBR between the tumors and the metastatic LNs.

### 3.2. Characteristics of Patients

Fifteen patients (thirteen with lung cancer and two with esophageal cancer) were enrolled in the clinical study (Table 1), comprising seven men and eight women, with a mean age of 66 ± 10 years (range, 49–84 years). No intraoperative or postoperative adverse event related to the ICG injection was observed. 

The surgical procedures performed on ten of the patients with primary lung cancer were VATS lobectomy in seven, robotic lobectomy in two, and VATS wedge resection in one. According to pathological analyses, eight patients in the lung cancer group were diagnosed with adenocarcinoma and two with squamous cell carcinoma. The mean cancer diameter was 2.4 ± 1.0cm (range: 0.7–3.9cm). The pathological TNM stages were as follows: four patients with T1N0M0, two with T2N0M0, one with T1N1M0, and three with T2N1M0. In total, 162 LNs were dissected in patients with lung cancer (mean: 16 [range: 1–31] LNs). Three patients with metastatic lung cancer from colorectal adenocarcinoma underwent VATS wedge resection. The mean cancer diameter was 1.1 ± 0.3 cm (range: 0.8–1.5 cm), and a total of five LNs were dissected (mean: two [range: one to three] LNs). 

Two patients with esophageal cancer underwent VATS esophagectomy and esophagogastrostomy. According to pathological analyses, both patients had squamous cell carcinoma, with cancer diameters of 3.7 cm and 10 cm, respectively. The pathological TNM stages were T2N0M0 and T3N2M0. In total, 98 LNs were dissected in patients with esophageal cancer (mean: 49 [range: 37–61] LNs).

### 3.3. Identification of Tumors and Metastatic LNs Using ICG Fluorescence Imaging in the Patients

We examined ICG fluorescence imaging for 15 tumors, and the images successfully identified all tumors. Among two hundred and sixty-five resected LNs from the fifteen patients, ICG fluorescence signals were analyzed in fifty-four LNs with suspected metastasis, including LNs with positive findings in the preoperative diagnostic tests (CT and PET/CT), lobe-specific LNs of lung cancer, anatomically key regional LNs of esophageal cancer, and enlarged LNs of metastatic lung cancer during surgery; eleven LNs were pathologically confirmed to be metastatic. Among the fifty-four LNs, twenty were detected on ICG fluorescence imaging, of which nine were false positives (45.0%) and eleven were true positives (55.0%). Overall, thirty-four LNs without fluorescent signals were pathologically confirmed to be nonmetastatic. NIR fluorescence imaging, shown in Figure 2, showed stronger fluorescence signals in tumors and metastatic LNs than in nonmetastatic LNs; the SBRs of the tumors (2.8 ± 0.4) and the metastatic LNs (2.6 ± 0.2) were significantly higher than that of nonmetastatic LNs (1.4 ± 0.2) (*p* < 0.0001 and *p* < 0.001, respectively). There were no significant differences in SBR between the tumors and the metastatic LNs. The sensitivity, specificity, accuracy, PPV, and NPV of the ICG fluorescence signal in the diagnosis of metastatic LNs were determined to be 100%, 79.1%, 83.3%, 55%, and 100%, respectively.

### 3.4. Comparison of Metastatic LN Detection Efficiency between PET/CT and NIR Fluorescence Imaging

We compared the diagnostic results of the ICG fluorescence imaging with those of the CT and the PET/CT to evaluate the clinical feasibility of intravenous ICG injection for metastatic LN detection (Figure 3). Among the fifty-four LNs, twelve metastatic LNs were suspected in the preoperative CT and PET/CT. The pathological results confirmed that of the twelve LNs detected using CT, five were true-positive, seven were false-positive, and six were false-negative; of the 12 LNs detected using PET/CT, six were true-positive, six were false-positive, and five were false-negative. The number of metastatic LNs detected using ICG fluorescence imaging was eleven, which was six and five more than those detected using CT and PET/CT, respectively. The number of false-positive LNs detected using ICG fluorescence imaging was nine, which was consistent with the seven and six detected using CT and PET/CT, respectively. None of the metastatic LNs was diagnosed as false-negative on ICG fluorescence imaging. The sensitivity, specificity, accuracy, PPV, and NPV of the CT were 45.5%, 83.7%, 75.9%, 41.7%, and 85.7%, respectively, whereas those of the PET/CT were 50%, 85.7%, 77.8%, 50%, and 85.7%, respectively.

## 4. Discussion

LN metastasis can lead to poor prognosis in lung or esophageal cancer. Therefore, LN dissection has been recognized as a standard treatment procedure for thoracic cancers, as it is necessary to improve the survival rate thereof [2,3,4,5]. However, considering that the median age at diagnosis of thoracic cancer is 70 years or older, these patients are usually affected by extensive comorbidities, such as cardiac and pulmonary function limitations [37,38], so extensive lymphadenectomy may not be the most suitable option. Therefore, accurate identification and removal of metastatic LNs may improve the quality of life of these elderly patients.

To the best of our knowledge, this is the first study to evaluate the feasibility of metastatic LN detection using ICG fluorescence imaging. Notably, we found that metastatic LNs that were previously undetectable on preoperative CT or PET/CT could be detected using ICG fluorescence imaging. We also found no metastasis in the absence of fluorescence signals in histopathology. Overall, ICG fluorescence imaging can provide accurate intraoperative information, thereby preventing missed diagnoses of metastasis.

For metastatic LN detection, SLN biopsy, which is endoscopically injecting ICG submucosally around a tumor, is considered the standard surgical treatment for certain cancers [39]. Compared with SLN biopsy, intravenous ICG may be beneficial for increased sensitivity but is less specific, resulting in more lymph nodes harvested and increased morbidity. For metastatic LN detection using intravenous ICG, the efficiency of this technique should be compared with SLN biopsy in future studies. In preclinical studies, we found that the SBRs of tumors and metastatic LNs were significantly higher than those of normal tissues and LNs. We also demonstrated that ICG was distributed in footpad tumors and metastatic LNs with GFP signals (Figure 1). ICG-based tumor-detection methods have been used for detection of many types of tumors [33,35,40]. 

ICG-based metastatic LN detection was performed in patients with tumors to evaluate the clinical feasibility of this method. In primary lung cancer, except for one patient who underwent wedge resection due to pulmonary dysfunction, all patients underwent lobectomy to improve resection of occult metastatic LNs that often occur in lung cancer [13]. The mean number of dissected LNs in nine patients was 16 (range, 9–31 LNs), which was similar to that proposed in other studies to provide an opportunity to identify positive LNs [41]. 

LN dissection is not a routine procedure for metastatic lung cancer [19]. However, the incidence of LN metastasis in this cancer ranges from 12 to 19%, and the negative impact of LN metastasis on survival has been demonstrated [17]. Adequate LN dissection has also recently been advocated for all metastatic lung cancers [18]. However, most researchers do not recommend routine lymphadenectomy unless LNs are enlarged, centrally located or require anatomic resection or LN involvement is suspected in preoperative examination [20]. In this study, we performed five LN dissections on three patients who underwent wedge resection, of which three had LNs resected because of intraoperative findings of enlargement and the remaining two were suspected of involvement during preoperative CT.

Esophageal cancer also frequently has occult LN metastasis [12]; in the present study, the mean number of dissected LNs was 49 (range: 37–61 LNs), which was sufficient for accurate N classification and to avoid missing occult metastatic LNs [11]. 

ICG fluoroscopic evaluation was performed for all 15 tumors and 54 LNs, indicating LNs with positive findings of lung and esophageal cancer in the preoperative diagnostic tests (CT and PET/CT), lobe-specific LNs of lung cancer, and enlarged LNs of metastatic lung cancer during surgery. Consistent with what was observed in mice, the SBRs of the tumors and the metastatic LNs were significantly higher than those of normal tissues and LNs, and there were no significant differences between the tumors and the metastatic LNs. Based on the difference in SBR between metastatic and normal LNs, we successfully detected all 15 tumors and eleven metastatic LNs (Figure 2). In fact, in a total of eleven metastatic LNs, four LNs were detected in four patients with lung cancer, three LNs in one patient with metastatic lung cancer, and four LNs in one patient with esophageal cancer. Collectively, if the ICG-based metastatic LN detection method developed in this study is applied to lung and esophageal cancer surgery in the future, we can expect to not only avoid missing occult metastatic LNs but also avoid unnecessary LN dissections. 

Furthermore, we compared the metastatic LN detection efficiencies of CT, PET/CT, and NIR fluorescence imaging for 54 LNs. Among the twenty LNs identified with NIR fluorescence imaging, nine were false positives (45.0%) and eleven were true positives (55.0%), with an accuracy of 83.3%. Six of the nine false-positive results were consistent with the CT and PET/CT imaging, and histopathology confirmed that the LNs were enlarged with inflammation; for the other three false-positive results, histopathology confirmed the absence of LN enlargement but the presence of inflammation. This result is consistent with those reported in several studies and shows that it can be challenging to distinguish a tumor from inflammation using ICG fluorescence imaging. Interestingly, the ICG fluorescence was able to detect six and five more metastatic LNs that were undetected in preoperative CT and PET/CT, respectively. Importantly, histopathology confirmed that the absence of metastasis in 34 LNs did not show a fluorescence signal, resulting in an ICG fluorescence-imaging sensitivity of 100%. The 100% sensitivity suggests that without ICG fluorescence imaging, LNs can be diagnosed as normal.

Although we successfully detected tumors and metastatic LNs using ICG fluorescence imaging, this study had several limitations. First, we only obtained fluorescence images of patient specimens postsurgery. Thus, it is necessary to verify the feasibility of ICG fluorescence imaging to detect metastatic LNs during surgery. Second, this study included a relatively small number of patients, including two patients with esophageal cancer. Therefore, it is critical to evaluate the metastatic LN detection efficiency of ICG via using a larger sample size. Finally, as ICG is not a targeting agent for tumors or metastatic LNs, and as it has relatively low fluorescence that makes it difficult to use to detect tumors in vivo, further research is needed to develop target contrast agents with high fluorescence signals to detect metastatic LNs.

## 5. Conclusions

In conclusion, we successfully visualized tumors and metastatic LNs simultaneously in mouse and human specimens using ICG fluorescence imaging. In a clinical study, we detected tumors and metastatic LNs with an accuracy of 83.3%. Importantly, ICG fluorescence imaging can detect metastatic LNs that were previously undetected during preoperative CT or PET/CT. Collectively, these findings suggest that ICG fluorescence imaging can accurately identify tumors and suspicious metastatic LNs intraoperatively.

## Figures and Tables

**Figure 1 cancers-15-01964-f001:**
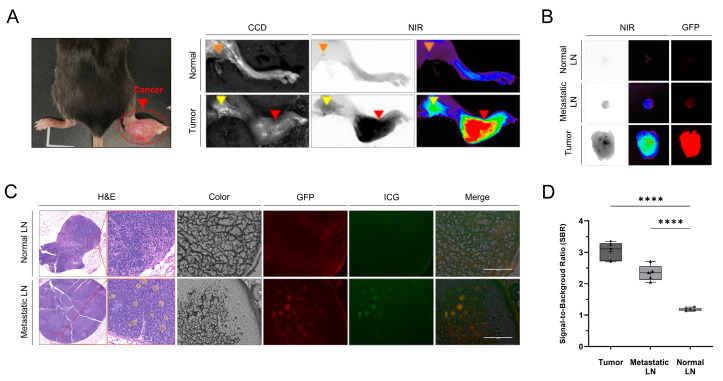
Identification of tumors and metastatic LNs using ICG fluorescence images in a mouse tumor model. (**A**) Representative CCD and NIR fluorescence images of normal footpad and popliteal LNs in a normal mouse and footpad tumor model with LN metastasis after intravenous injection of ICG (5 mg/kg). Red arrows indicate cancer, yellow arrows indicate metastatic LNs, and orange arrows indicate normal LNs. (**B**) Ex vivo fluorescence images of a tumor, metastatic LNs, and normal LNs excised 12 h after ICG injection. (**C**) Representative H&E staining and CCD, GFP [1], NIR (green), and merged (CCD, GFP, and NIR) microscopic images of normal LNs and metastatic LNs (scale bar: 100 μm). (**D**) Comparison of the signal-to-background ratios of the tumor, the metastatic LNs, and the normal LNs 12 h after ICG injection (n = 4; statistical analysis was performed using one-way analysis of variance followed by Tukey’s multiple comparison test; tumors vs. normal LNs, **** *p* < 0.0001; metastatic LNs vs. normal LNs, **** *p* < 0.0001). LN, lymph node; NIR, near-infrared; GFP, green fluorescent protein; ICG, indocyanine green; H&E, hematoxylin and eosin; CCD, charge-coupled device.

**Figure 2 cancers-15-01964-f002:**
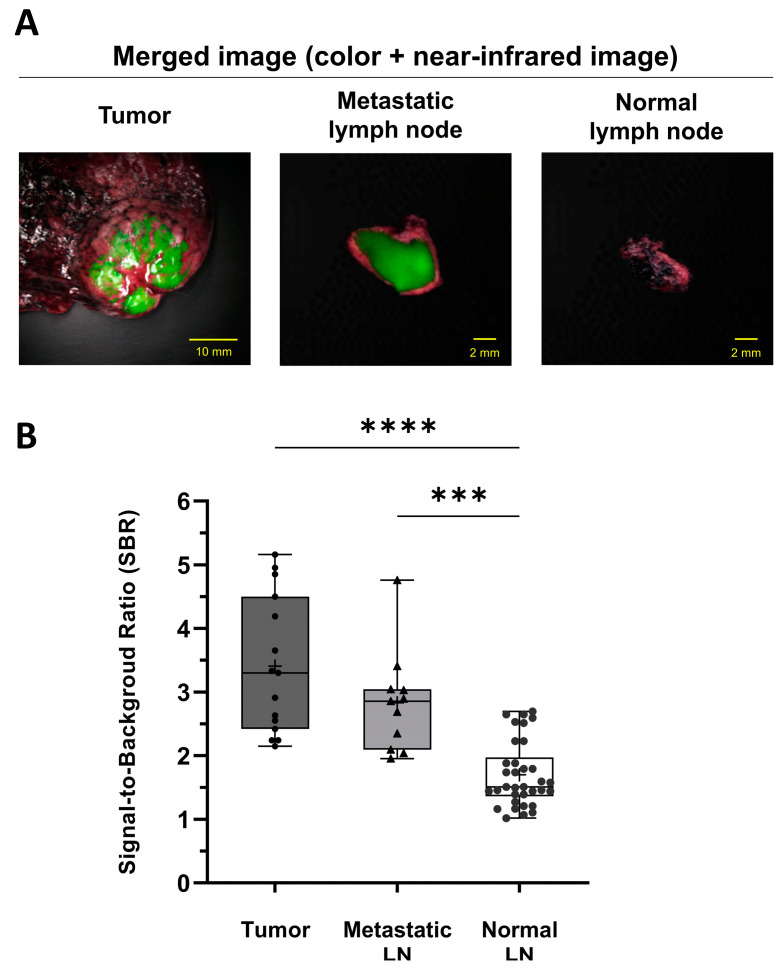
Identification of tumors and metastatic LNs using ICG fluorescence imaging in a patient specimen. (**A**) Representative ex vivo fluorescence images of tumors, metastatic LNs, and normal LNs after intravenous injection of ICG (2 mg/kg). (**B**) Signal-to-background ratios of the tumors, metastatic LNs, and normal LNs (n = 5; statistical analysis was performed using one-way analysis of variance followed by Tukey’s multiple comparison test; tumors vs. normal LNs, **** *p* < 0.0001; metastatic LNs vs. normal LNs, *** *p* < 0.001). LN, lymph node; ICG, indocyanine green.

**Figure 3 cancers-15-01964-f003:**
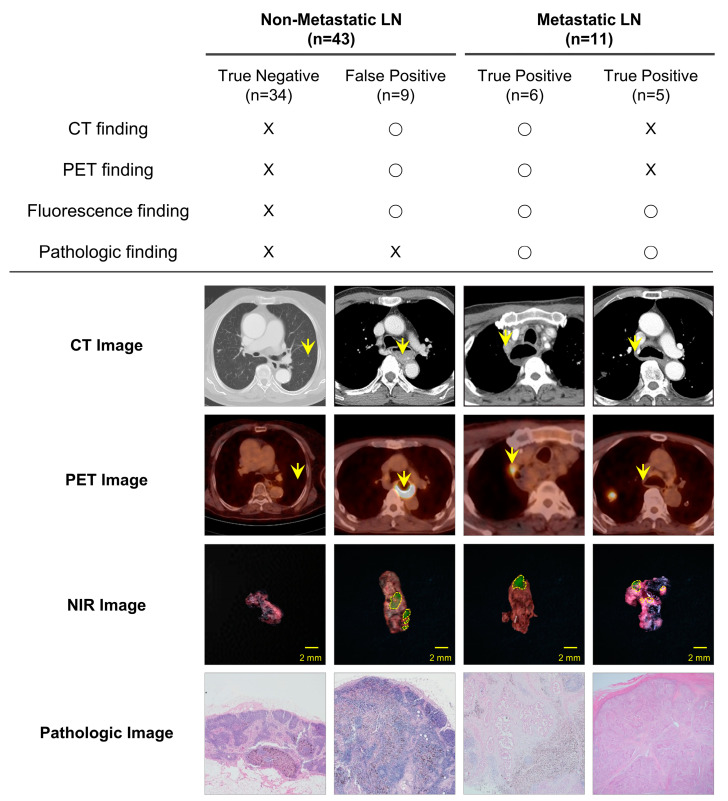
Comparison of the detection efficiencies of CT, PET/CT, and ICG imaging for LNs in patients. Representative CT, PET/CT, NIR (green) fluorescence, and pathological images of patients. The patients were classified into two groups (nonmetastatic and metastatic LNs) based on the pathologic results. The yellow arrows indicate metastatic LNs, “O” indicates the presence of metastasis, and “X” indicates the absence of metastasis. CT, computed tomography; PET/CT, positron-emission tomography/computed tomography; NIR, near-infrared; LN, lymph node.

**Table 1 cancers-15-01964-t001:** Characteristics of patients with thoracic cancer.

Variable	Data (n = 15)
Median Age (Years)	
Mean ± SD (Range)	66 ± 10 (49–84)
Gender	
Female	8 (54%)
Male	7 (46%)
Cancer Type	
Lung Cancer (n = 10)	
Surgery	
Lobectomy	9 (90%)
Wedge Resection	1 (10%)
Histological Type	
Squamous Cell Carcinoma	2 (20%)
Adenocarcinoma	8 (80%)
TNM Stage	
T1N0M0	4 (40%)
T2N0M0	2 (20%)
T1N1M0	1 (10%)
T2N1M0	3 (30%)
Dissected Lymph Nodes	
Mean (Range)	16 (1–31)
Metastatic Lung Cancer (n = 3)	
Surgery	
Wedge Resection	3 (100%)
Histological Type	
Adenocarcinoma from Colon Cancer	3 (100%)
Dissected Lymph Nodes	
Mean (Range)	2 (1–3)
Esophageal Cancer (n = 2)	
Surgery	
Esophagectomy	2 (100%)
Histological Type	
Squamous Cell Carcinoma	2 (100%)
TNM Stage	
T2N0M0	1 (50%)
T3N2M0	1 (50%)
Dissected Lymph Nodes	
Mean (Range)	49 (37–61)

## Data Availability

The data presented in this study are available upon request from the corresponding authors.

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
