# Peer review of "Identification of Metastatic Lymph Nodes Using Indocyanine Green Fluorescence Imaging"

_cancers, 2023, doi:10.3390/cancers15071964_

Round 1
Reviewer 1 Report
Overall nice work with the mouse model and further investigation into clinical samples.
Strengths:
1. Visually appealing figures. Data is shown in box plots.
2. It reads reasonably well in English with some minor proofreading and modifications required.
3. The research group is capable of translational research with basic and clinical infrastructure established.
However, there are a few major questions or concerns:
1. The authors make multiple claims that to their knowledge this is the first study investigating ICG in LN mets for lung and/or esophageal cancer. However, there have been at least two systematic reviews and meta-analyses of this technique. See Sun et al. 2020 (DOI: 10.1007/s11748-020-01400-8.) and Jimenez-Lillo et al. 2021 (DOI: 10.1245/s10434-021-09617-4.) for publications in lung and esophageal cancer, respectively. At a minimum, the author's claims should be stepped back and an additional literature review should be performed to include these papers and any others they identify.
2. The methods require revision:
a. How was the normal model established? The sentence in line 132-133, "Ten mice with LN metastasis were divided into two groups: 132 normal model (n = 5) and footpad tumor model (n = 5)." confuses the reader. Why does the normal group have lymph node metastasis? Is this cancer cell line without GFP? This reviewer infers that the normal model does not have any cancer cells present and is essentially a normal control. However, this should be more clearly stated. Furthermore, if the normal model does not have any injection of Lewis lung cancer cells, is it possible that GFP is a confounder in identifying LN mets with NIR in the mouse model? If there has been no negative control established for footpad tumours, there should be justification within the text (ie. prior evidence that NIR cannot excite GFP or that GFP cannot interact with ICG dye).
b. How was the histology and fluorescent imaging obtained? Is this fresh frozen HCT or formalin fixed paraffin embedded tissue?
c. The reader should be informed as to what a charged-coupled device is (or at least a reference provided). What does it visualize? Why is it relevant?
d. Statistical analysis: what is the justification for the authors using one-way ANOVA in the mouse model data versus a non-parametric approach given the small sample size?
3. Figure 3 - Ensure consistent capitalization of table headings (ex. True Negative in one heading then True positive in the next)
4. Line 62 - need to add 'a' to "lung is common site of malignant tumour..."
5. Paragraph 2 = too many conjunctive adverbs to start sentences. The sentences all start with one of, "However, Thus, Conversely and Consequently". Please revise.
6. Line 341 - missing 'that'.
Please proofread the document for additional missing words/grammar throughout.
Author Response
We thank the reviewer for the valuable comment.
"Please see the attachment.

Reviewer 2 Report
This manuscript describes a study assessing whether ICG fluorescence imaging is able to detect metastatic lymph nodes, specifically for lung cancer and esophageal cancer patients. The writing is clear and most of the scientific design is reasonable. I have a problem with the rationale of the study though. Currently the manuscript is written as though ICG fluoroscopy is a method that can ‘detect cancer’, however ICG is not a targeting agent and has no real mechanism to confirm that a lesion with high fluorescence is a tumour until pathology is subsequently carried out. Similarly, ICG will not be able to confirm that a LN is metastatic until pathology is performed on the LN with high fluorescence. There is only one sentence about this at the very end of the Discussion. I recommend the authors edit their Abstract, Summary and Introduction to clarify this important detail earlier on.
My additional comments are as follows:
1) When explaining that ICG has been used to detect ‘cancers’, I think this term would better be stated as ‘tumours’ throughout the manuscript
2) There are some overstatements, which I think need to be softened. Specifically, the statement in the Abstract which says that ‘ICG can be used to detect metastatic LNs and cancer using fluorescence imaging’ (refer to my comments above) and the last sentence of the Conclusions which is inferring the results here will translate into intraoperative use, which has not yet been proven.
3) Reference #34 is not a good reference to use to support the statement that ICG can be used to detect ‘several types of tumours’ as their conclusions states the opposite, i.e. that “ICG fluorescence pattern is … unable to reliably predict malignant vs benign lesions”.
4) The Methods could be streamlined, I suggest moving Section 2.1 which describes the ICG procedure within sections 2.4 and 2.5 so it is clear how ICG was prepared for the animal model and patients respectively. Currently it is confusing explaining this preparation prior to any of the animal and patient study methods.
5) The Methods would benefit from more information about the NIR imaging, what was the ‘NIR fluorescence imaging system’ used? What was the resolution? How long was it performed for, and was there a systematic distance between the animal/patient and the camera, to ensure there wasn’t a large variability in fluorescence intensities, for example? To aid this clarify, I would suggest adding a specific Section 2.7 heading covering the NIR imaging, which would start at the paragraph which begins ‘After resecting the cancer and LNs, ICG fluorescence imaging… etc’
6) Table 1 is very long and challenging to read. Is there a better way to present this information?
7) Figure 3, I found this confusing to understand. Specifically, the table with + and – within it. Can this be simplified to aid understanding?
Author Response

(The authors gave the same response as above.)

Round 2
Reviewer 1 Report
Dear authors,
Thank you for your dedication to your research and your responses. Overall the majority of my comments have been satisfied. I still believe a few minor tweaks should be included before publication, but ultimately, the editor may think otherwise.
Responses to author comments:
Comment 1 - re: intravenous versus mucosal injection and citation of relevant literature
At the very least it is incredibly notable that significant work of an alternative technique has been completed. This paper should outline the pros and cons of your technique compared to previous methods and should argue at least equivalent performance to be notable (or, at least a justification as to why your technique warrants publication). For example, it appears that intravenous injection compared to endoscopic mucosal injection may favour an improved sensitivity but worse specificity. This trade-off could be an advantageous argument for the authors. Given this, however, perhaps the worse specificity would contribute to more nodes harvested and thus more morbidity. I anticipate the authors agree that maximizing sensitivity for this application is incredibly valuable to ensure no true metastatic nodes remain in-situ.
Overall, this does not need to be exhaustive but at least a sentence or two should highlight this debate.
Comment 2 - re: confusing wording for methods section and GFP/ICG imaging wavelengths.
Thank you for the clarification! The method here is now clear.
Also, thanks for your informative comments regarding imaging wavelengths. In this reviewer’s opinion, the imaged wavelengths are notable and should be included unless the authors can provide a reason for them not to be. Published literature has demonstrated that ICG is excitable at a range of wavelengths (doi: 10.1002/jso.21943) so for reproducibility (and notable difference in absorption between ICG and GFP) this information should be included.
All other additional comments are satisfactory and no further changes are suggested.
